# Microbiota Composition of Mucosa and Interactions between the Microbes of the Different Gut Segments Could Be a Factor to Modulate the Growth Rate of Broiler Chickens

**DOI:** 10.3390/ani12101296

**Published:** 2022-05-18

**Authors:** Valéria Farkas, Gábor Csitári, László Menyhárt, Nikoletta Such, László Pál, Ferenc Husvéth, Mohamed Ali Rawash, Ákos Mezőlaki, Károly Dublecz

**Affiliations:** 1Institute of Physiology and Nutrition, Department of Animal Nutrition and Nutritional Physiology, Georgikon Campus, Deák Ferenc Street 16, Hungarian University of Agriculture and Life Sciences, 8360 Keszthely, Hungary; farkas.valeria@uni-mate.hu (V.F.); csitari.gabor@uni-mate.hu (G.C.); such.nikoletta.amanda@uni-mate.hu (N.S.); pal.laszlo@uni-mate.hu (L.P.); husveth.ferenc@uni-mate.hu (F.H.); mohamedali@post.agr.cu.edu.eg (M.A.R.); 2Institute of Mathematics and Basic Science, Georgikon Campus, Deák Ferenc Street 16, Hungarian University of Agriculture and Life Sciences, 8360 Keszthely, Hungary; menyhart.laszlo@uni-mate.hu; 3Agrofeed Ltd., Duna Kapu Square 10, 9022 Győr, Hungary; akos.mezolaki@agrofeed.hu

**Keywords:** microbiota, body weight, jejunum mucosa, caecum chymus, correlation, co-occurrence relationships

## Abstract

**Simple Summary:**

The microbial communities inhabiting the gastrointestinal tract (GIT) of chickens are essential for the gut homeostasis, metabolism, and health status of the host animal. Previous studies exploring the relationship between chicken growth performance and gut microbiota focused mostly on gut content and excreta samples, neglecting the mucosa that promotes colonisation by distinct groups of microorganisms. These observations emphasised the importance of studying the variations between the bacterial communities of the lumen and mucosa throughout the different sections of the GIT. The novelty of this study is that we have evaluated the microbial communities of the jejunum chymus, jejunum mucosa, and caecum chymus of broiler chickens with different growth rates. Besides the bacteriota composition, the interactions between the bacteria were also evaluated. We have confirmed that the microbiota composition is influenced mostly by the sampling place. However, some body weight (BW)-related changes and interactions have also been found. In these cases, the mucosa seems to play a crucial role.

**Abstract:**

The study reported here aimed to determine whether correlations can be found between the intestinal segment-related microbiota composition and the different growing intensities of broiler chickens. The bacterial community structures of three intestinal segments (jejunum chymus—JC, jejunum mucosa—JM, caecum chymus—CC) from broiler chickens with low body weight (LBW) and high body weight (HBW) were investigated. Similar to the previous results in most cases, significant differences were found in the bacteriota diversity and composition between the different sampling places. However, fewer body weight (BW)-related differences were detected. In the JM of the HBW birds, the *Bacteroidetes*/*Firmicutes* ratio (B/F) was also higher. At the genus level significant differences were observed between the BW groups in the relative abundance of *Enterococcus*, mainly in the JC; *Bacteroides* and *Ruminococcaceae UCG-010*, mainly in the JM; and *Ruminococcaceae UCG-013*, *Negativibacillus*, and *Alistipes* in the CC. These genera and others (e.g., *Parabacteroides* and *Fournierella* in the JM; *Butyricoccus*, *Ruminiclostridium*-*9*, and *Bilophila* in the CC) showed a close correlation with BW. The co-occurrence interaction results in the JC revealed a correlation between the genera of *Actinobacteria* (mainly with *Corynebacterium*) and *Firmicutes Bacilli* classes with different patterns in the two BW groups. In the JM of LBW birds, two co-occurring communities were found that were not identifiable in HBW chickens and their members belonged to the families of *Ruminococcaceae* and *Lachnospiraceae*. In the frame of the co-occurrence evaluation between the jejunal content and mucosa, the two genera (*Trichococcus* and *Oligella)* in the JC were found to have a significant positive correlation with other genera of the JM only in LBW chickens.

## 1. Introduction

In recent decades, breeding programs have improved the efficiency of poultry production tremendously. The parameters influencing the performance and efficiency of chicken flocks, such as feed conversion rate (FCR) and body weight gain (BWG), are critical factors for broiler producers. Among other factors gut health also plays an important role as a modulatory factor of the production traits.

The modulation of the gastrointestinal tract (GIT) microbiota (e.g., addition of pre- or probiotics, feed additives) is used widely in poultry production. Several studies have attempted to identify how the intestinal microbes are associated with the production traits (e.g., weight gain, fed conversion ratio) [1,2,3,4]. However, the findings thus far have been inconsistent and sometimes contradictory [5]. Some investigations focusing on the relationship between gut microbiota and obesity suggested that the gut microbiota affects energy utilisation and energy deposition of humans and animals [6,7,8]. Several authors revealed a positive correlation between the increased *Firmicutes*/*Bacteroidetes* (F/B) ratio in the colon or caecum and weight gain or obesity [7,9,10]. In opposition to these results, a number of studies did not observe any modifications of this parameter, and even a decreased F/B ratio in obese animals and humans has been found [9,10,11,12]. The F/B ratio can vary significantly due to the age, nutrition, and health status of the host. The relationship between the F/B ratio and obesity is still the subject of intensive research. Different bacterial diversity at different taxonomy levels in obese/lean animals suggests that a more detailed evaluation of bacteriota composition is more relevant than only the F/B ratio [13]. Whole genome sequencing showed that representatives of *Bacteroidetes* and *Firmicutes* have the genetic potential to follow different strategies of gut colonisation, which may explain their coexistence in chicken caecum [14]. Members of the phyla *Firmicutes* and *Bacteroidetes* also play an important role in the intestinal mucosa, but only a few results are available in this area.

Gut microbes colonise the outer layer of the mucus and use nutrients from the mucus itself. A recent comparison between lumen and mucosa associated microorganisms revealed a much greater microbial community richness in the mucosa, particularly in the ileum and caecum of broiler chickens [15].

An emerging approach is to study the so-called correlation networks between different bacteria. One possibility to infer interactions between bacteria in the gut microbiota is to quantify the co-occurrence of operational taxonomic units (OTUs) across multiple samples [16].

The aim of this study was to determine potential correlations between the intestinal segment-related microbiota and the different growing intensities of broiler chickens. Furthermore, possible interactions between the microbiota of certain gut segments were also investigated with a co-occurrence network analysis.

## 2. Materials and Methods

### 2.1. Ethics and Approval Statement

The trial was carried out at the experimental farm of the Institute of Physiology and Nutrition, Hungarian University of Agriculture and Life Sciences (Georgikon Campus, Keszthely, Hungary). All husbandry and euthanasia procedures were performed in accordance with the Hungarian Government Decree 40/2013 and in full consideration of animal welfare ethics. The animal experiment was approved by the Institutional Ethics Committee (Animal Welfare Committee, Georgikon Campus, Hungarian University of Agriculture and Life Sciences) under the license number MÁB-1/2017.

### 2.2. Animal Handling and Sampling

A total of 160 male broiler chicks (Ross 308) obtained from a commercial hatchery (Gallus Ltd., Devecser, Hungary) were raised in a chopped wheat-straw floor pen. The initial mean body weight (BW) of the chicks was 45.4 ± 0.36 g. Commercial wheat-, corn-, and soybean-based broiler starter (d 1–11), grower (d 12–25), and finisher (d 26–37) diets were fed ad libitum. The composition and nutrient content of the diets can be found in Appendix A. The environmental conditions (heating, lighting, and ventilation) were in accordance with the breeder’s instructions [17]. On day 37, all chickens were weighed individually and the average BW (mean ± standard deviation (SD)) was determined. The average BW of the flock was 2696 ± 227 g. The live weight of the birds and the variation coefficient of the flock corresponded to the Ross 308 broiler performance objectives of 37-day-old male chickens [17]. Afterwards, two groups of chickens were defined. Out of the group of the lower body weight (LBW) chicks, 32 birds were selected with a live weight below the average minus the standard deviation (SD) of the whole population. Similarly, 26 chickens of the higher body weight (HBW) treatment were selected from the animals having a live weight above the average plus SD. All selected birds were healthy, without signs of disease or leg problems. The live weight distribution of the randomly selected chickens was 2235 ± 76 g and 3098 ± 173 g in the LBW and HBW treatments, respectively (Appendix A).

Five birds from each group were selected randomly and euthanised with CO_2_ and slaughtered. The digestive tract was removed immediately, and gut contents and mucosa samples collected. The jejunal chyme was taken from a 10 cm long gut segment, directly proximal to Meckel’s diverticulum. Caecal contents were collected from the left caecal sack. The chyme of the jejunum and the caecal content was gently pushed out. After the gut content collection, the jejunal part was washed with sterile ice-cold phosphate buffer solution (PBS) until the mucosa was completely cleaned from the digesta. Mucosa samples were collected aseptically by scraping off the mucosa from the internal wall of the gut with a glass slide. All samples were homogenised and stored at −80 °C until further processing occurred.

### 2.3. DNA Extraction, 16S rRNA Gene Amplification and Illumina MiSeq Sequencing

Bacterial DNA was extracted from 15 mg samples using the AquaGenomic Kit (MoBiTec GmbH, Göttingen, Germany) and further purified using KAPA Pure Beads (Roche, Basel, Switzerland) according to the manufacturer’s protocols. The concentration of genomic DNA was measured using a Qubit 3.0 Fluorometer with the Qubit dsDNA HS Assay Kit (Thermo Fisher Scientific Inc., Waltham, MA, USA). Bacterial DNA was amplified with tagged primers (forward, 5′TCGTCGGCAGCGTCAGATGTGTATAAGAGACAGCCTACGGGNGGCWGCAG, and reverse, 5′GTCTCGTGGGCTCGGAGATGTGTATAAGAGACAGGACTACHVGGGTATCTAATCC) covering the V3–V4 region of the bacterial 16S rRNA gene [18]. Polymerase chain reactions (PCRs) and DNA purifications were performed according to Illumina’s demonstrated protocol (Illumina Inc., 2013). The PCR product libraries were quantified and qualified by using the High Sensitivity D1000 ScreenTape on the TapeStation 2200 instrument (Agilent Technologies, Santa Clara, CA, USA). Equimolar concentrations of libraries were pooled and sequenced on an Illumina MiSeq platform using a MiSeq Reagent Kit v3 (600 cycle; Illumina Inc., San Diego, CA, USA) with a 300-bp read length paired-end protocol. Raw sequences data of 16S rRNA gene analysis were deposited at the National Center for Biotechnology Information (NCBI) Sequence Read Archive (SRA) under accession number PRJNA609272.

### 2.4. Bioinformatics and Statistical Analyses

Microbiome bioinformatics were performed with the Quantitative Insights Into Microbial Ecology 2 (QIIME2) version 2020.2 software package [19]. Raw sequence data were demultiplexed and quality filtered using the q2-demux plugin, followed by denoising with Deblur [20]. Sequences were filtered based on quality scores and the presence of ambiguous base calls using the quality-filter q-score options (QIIME2 default setting). Representative sequences were found using a 16S reference as a positive filter, as implemented in the Deblur denoise-16S method. Sequences were clustered into operational taxonomic units (OTUs) using VSEARCH algorithm open-reference clustering, based on a 97% similarity to the SILVA (release 132) reference database [21]. Alpha diversity metrics (Chao1, Shannon, Simpson, and phylogenetic distance(PD)) and beta diversity metrics (Bray–Curtis dissimilarity) were estimated using the QIIME2 diversity plugin and MicrobiomeAnalyst (https://www.microbiomeanalyst.ca/, accessed on 1 September 2020) online software after samples were rarefied to 10,000 sequences per sample [22]. Only features appearing in at least a minimum of 10 reads across all samples were retained in the resulting feature table. To examine differences in microbial community structures between samples, a principal coordinate analysis (PCoA) with the Bray–Curtis dissimilarity was generated using the MicrobiomeAnalyst online software. Permutational multivariate analysis of variance (PERMANOVA, *p* < 0.05) was used to analyse spatial variation in beta diversity and the effects of sampling places (JC, JM, and CC) and BW. Canonical correspondence analysis (CCA) was used to evaluate the effect of BW on the intestinal tract microbiota structure. Correlations of the canonical axes with the explanatory matrix were reported and the significance of each correlation was determined by 999 permutations with Calypso online software [23].

Statistical analysis was performed with SPSS statistical software version 23.0 (IBM Corp. Released 2015) and Calypso. Alpha diversity indices and the microbial composition at different taxonomical levels and in different intestinal sampling places (JC, JM and CC) were compared using a two-way ANOVA test with Tukey’s HSD multiple group comparison post hoc test, using the sampling places (SPs) and the BW of chickens (LBW, HBW) as main factors. Although microbial abundance data follow non-normal distribution, the hypothesis testing was conducted by ANOVA rather than its nonparametric counterpart. ANOVA is more powerful to test the interaction and the literature confirmed its robustness to non-normality, especially in cases of equal sample sizes and similar distributional shapes as was valid for our data [24]. The Benjamini–Hochberg false discovery rate (BH-FDR) correction (FDR *p*-value) was used to adjust for multiple testing. Statistical significance was defined as the FDR of *p* < 0.05, whereas the FDR *p*-value between 0.05 and 0.10 was considered as a trend. A Venn diagram was made with the InteractiVenn web-based tool [25]. Correlations between gut microbial composition (relative abundance of genera) and the BW-related change at the different sampling places were evaluated with Spearman’s correlation. The correlations with a *p*-value less than 0.05 was considered as statistically significant. The analysis was performed between each SP and BW category for all genera representing more than 0.01% of the total microbial population and could be found in at least 8 samples.

### 2.5. Co-Occurrence Network Analysis

Differences in correlation structure between HBW and LBW chickens were analysed with Spearman’s rank correlation coefficients (rho). The correlation structures were revealed as follows. Only genera with a relative frequency greater than zero in at least four out of five samples were included. A Monte Carlo simulation was used to determine whether the number of correlation connections of a genus was significant or not. Spearman’s correlation was calculated for each pair of series and for each SP and BW combination.

Similar analysis was carried out to reveal the connection between the microbiota composition of the jejunal content and jejunal mucosa.

## 3. Results

### 3.1. Sequencing Data and Differences in the Alpha and Beta Diversity

In total, 582,263 quality-controlled sequences were generated with a mean of 208,595 (SD: 4306), 166,088 (SD: 3927), and 207,580 (SD: 3039) reads per JC, JM, and CC samples, respectively. Sequences were classified into 826 OTUs. The efficiency of classification at different taxonomic levels was 80%, 98%, 99%, and 100% for the genus, family, order, and class and phylum, respectively. Rarefaction curves represented lower species richness in the JC and higher in JM and CC samples (Appendix A).

To describe which unique OTUs were present in the three sampling places, an OTU was assumed to be unique if it had at least one count in a given sample belonging to the sampling group. Among the three intestinal SPs, 210 (minimum core microbiota) shared OTUs were identified, whereas the aggregate OTU numbers were the largest in the JM (762 OTUs), and the lowest in the JC (381 OTUs; Appendix A). Shared OTUs among low and high BW chicken microbiota were 189, 526, and 505 in the JC, JM, and CC, respectively, which were more than 95% of the total reads number. The unique OTU number was mostly below 2%. The highest percentage was found in the jejunal mucosa of HBW birds (2.6%) (Appendix A).

All alpha diversity indices were significantly lower (*p* < 0.001) in the JC samples than in the JM and CC (Table 1). All measures (Chao1, Shannon, and Simpson) were the highest in the CC except PD, where the JM was more diverse. No significant BW effects and SP × BW-interactions were found (*p* > 0.05).

Beta-diversity was examined with the Bray–Curtis dissimilarity and canonical correspondence analysis (CCA) using the chi-square distance. In the case of a Bray–Curtis dissimilarity which gives equal weights to rare and abundant species, the sampling places were separated significantly (ANOSIM R = 0.741, *p* < 0.001; Figure 1A) but the BW groups were not (ANOSIM R = −0.01, *p* = 0.429; Figure 1B). In the case of a chi-square distance which gives more weight to rare species than to common species the separation by sampling places was even more pronounced (*p* = 0.001; Figure 1C) and in this case the separation by bodyweight was also significant (*p* = 0.045; Figure 1D).

### 3.2. Gut Microbiota Composition at Phylum Level

At the phylum level, the dominant phyla in the JC samples were *Firmicutes* (82.42%), *Proteobacteria* (6.83%), *Cyanobacteria* (4.29%), and *Actinobacteria* (3.47%) (Table 2). These four phyla represented more than 97% of the total bacterial population in the JC. In the JM and CC, the two dominant phyla were *Firmicutes* (64.77% and 49.35%, respectively), and *Bacteroidetes* (30.41% and 48.43%, respectively).

The relative abundance of *Firmicutes* (*p* < 0.001), *Proteobacteria* (*p* = 0.045), and *Bacteroidetes* (*p* < 0.001) was different across sampling places. *Firmicutes* represented significantly higher frequencies in the JC than in the JM and CC. *Proteobacteria* also represented significantly higher frequencies in the JC than in the JM and CC. On the other hand, the abundance of *Bacteroidetes* was the highest in the CC and JM.

Comparing the BW effects, *Bacteroidetes* represented a significantly higher abundance in HBW chickens (*p* = 0.029). In the jejunal mucosa both *Bacteroides* and *Tenericutes* represented a higher ratio in HBW birds. On the opposite hand, the abundance of *Firmicutes* in the JM was higher in the LBW group. Due to the higher abundance of *Bacteroidetes* in HBW birds, the *Bacteroidetes*/*Firmicutes* (B/F) ratio was also significantly higher in HBW chickens (*p* = 0.029). This difference was mainly the characteristic of the mucosa. No significant BW × SP interactions were found (*p* > 0.05).

### 3.3. Gut Microbiota Composition at Genus Level

At the genus level, 20 genera had about at least 1% relative abundance. The distribution of major bacterial genera is shown in Appendix A. *Lactobacillus* (65.1%), *Streptococcus* (7.2%), *Escherichia-Shigella* (4.6%), and *Corynebacterium-1* (2.8%) were dominant in the JC, whereas *Bacteroides* (26.7% and 44.3%), *Lactobacillus* (14.6% and 3.3%), *Ruminococcaceae UCG-014* (3.1% and 4.1%), and *Faecalibacterium* (2.8% and 2.2%) were dominant in the JM and CC, respectively.

The BW-dependent differences are summarised in Table 3. The relative abundance of *Enterococcus* in the JC and *Bacteroides* and *Ruminococcaceae UCG-010* in the JM were significantly higher in the HBW group. On the other hand, the abundance of *Ruminococcaceae UCG-013* and *Negativibacillus* in the CC was significantly higher in LBW chickens.

The BW × SP interaction showed significant effects in the case of *Enterococcus* (*p* = 0.001), *Ruminococcaceae* UCG-013 (*p* = 0.026), and *Negativibacillus* (*p* = 0.027). The reason for the interaction was the different trends of the three genera in the different sampling places. It is important to note that, although not significantly, *Lactobacillus* (LBW: 21.87%; HBW: 7.23%) showed a big difference in the JM.

### 3.4. Correlation between Gut Microbiota and BW

To identify specific microbial genera significantly associated with BW, Spearman’s rank correlation was performed. We performed the analysis between each sampling place of BW with genera representing >0.01% of the total microbial composition if presented in at least eight samples of birds (Appendix A).

In the JC, only *Enterococcus* (R = 0.87, *p* = 0.001) was positively correlated with BW (Appendix A). In the JM, six correlations were detected. Three of these, assigned to the phylum *Bacteroidetes* and class *Bacteroidia*, namely, *Parabacteroides* (R = 0.66, *p* = 0.044), *Alistipes* (R = 0.77, *p* = 0.01), and *Bacteroides* (R = 0.65, *p* = 0.049), correlated positively with the BW. In the phylum *Firmicutes*, *Clostridia* (class), *Fournierella* (R = 0.65, *p* = 0.049) and *Ruminococcaceae UCG-010* (R = 0.77, *p* = 0.009) were also positively correlated with BW (Appendix A). *CHKCI002* (genus of phylum *Actinobacteria*) was the only genus with negative significant correlation (R = −0.66, *p* = 0.038).

In the CC, only seven negative correlations were detected (Appendix A). Among them, six genera belonged to the *Clostridia* class (*Defluviitaleaceae UCG-011*, *GCA*-*900066575*, *Butyricicoccus*, *Negativibacillus*, *Ruminiclostridium*-*9,* and *Ruminococcaceae UCG-013*) and to the class Deltaproteobacteria (*Bilophila)*.

It was interesting that genera of the *Bacteroidetes* phylum (*Alistipes*, *Bacteroides,* and *Parabacteroides*) showed correlation mostly in the JM, whereas those of *Firmicutes* (*Butyricicoccus*, *Negativibacillus,* etc.) showed correlation in the CC.

### 3.5. Analysis of Co-Occurrence Patterns

The aim of this analysis was to infer interactions between bacteria genera in the gut microbiota across different BW groups and sampling places.

In the JC of LBW birds, six genera (*Corynebacterium, Corynebacterium-1, Dietzia, Globicatella, Lactococcus* and *Nosocomiicoccus*) had significant positive correlation (*p* < 0.001) (Figure 2A). In the HBW treatment group, four genera (*Corynebacterium-1, Brevibacterium, Aerococcus,* and *Jeotgalicoccus*) showed significant correlations (*p* = 0.013) (Figure 2B). In both cases, genera of the *Actinobacteria* phylum were associated with members of the *Bacilli* class of the *Firmicutes* phylum.

In the JM, only in the case of LBW chickens were significant correlations found among bacterial genera. Two such interaction matrixes have been detected. In the first network, seven genera (*Anaerofilum, Christensenellaceae R-7 group, Lachnoclostridium, Negativibacillus, Ruminiclostridium-5, Ruminococcus-1,* and *Sellimonas*) had positive correlations with each other (*p* = 0.011) (Figure 2C). The correlation between the other six genera (*Eubacterium coprostanoligenes group, Family XIII UCG-001, GCA-900066575, Ruminococcaceae NK4A214 group, Ruminococcaceae UCG-014,* and *Ruminococcus torques group*) in the second group (Figure 2D) was close to the significant level (*p* = 0.056). Similar to the first network, members of this group also included genera from the *Ruminococcaceae* and *Lachnospiraceae* families and the *Clostridiales* order.

In the case of JM-HBW, CC-LBW, and CC-HBW relations, no significant connections at the genus level were found.

Regarding the connection between the JC and JM, a significant number of correlations were found only in the LBW treatment group. *Trichococcus* from the JC had positive correlations with seven genera from the JM (*p* = 0.002; Figure 2E). *Oligella* from the JC was also positively correlated with six genera from the JM (*p* = 0.016; Figure 2F). All the correlations between the JC and JM were positive.

## 4. Discussion

The efficiency of poultry production can be measured with different parameters including the feed conversion ratio (FCR), body weight gain (BWG), or European efficiency index (EEI) [26]. The interaction between gut microbiota and production traits has been widely studied. However, the findings are sometimes contradictory or non-conclusive considering that the cause/effect relationship is still unclear. The factors that impact microbiota composition (age, breed, health status, and farm conditions including type of diet, feed additives, environment, and farm management) can also modify the fed intake and growth rate and may explain the variability in performance.

In our experiment, healthy chickens with the highest and lowest body weight (BW) were selected at day 37. The aim of the study was to determine whether correlations can be found between intestinal microbiota composition and the growing intensity of broiler chickens.

### 4.1. Diversity of Gut Microbiota

Microbial α diversities were significantly lower in the JC than in the JM and CC. Diversities were not significantly different between the JM and CC. Between the two BW groups no significant differences were found in the alpha diversity at any of the sampling places. Among the observed OTU numbers, the Shannon and Simpson indices tended to show greater diversity in the HBW group in the JM and CC. Our results are similar to those of Liu et al. who also failed to detect differences in the diversity of the ileum and caecal microbial community between chickens with different feed efficiency [27]. Some authors have reported lower species richness in the intestine of chickens with higher feed efficiency, but no differences if the excreta samples were analysed [28,29]. However, several studies found that bacterial diversity in the intestinal tract is higher in the birds with a lower FCR or high feed efficiency [1,2,3,4,26].

In previous studies with broiler chickens, the microbiota analysis focused mostly on gut content samples and less attention has been paid to the bacteria of the mucosa. Similar to our findings, in a recent comparison higher microbial community richness was found in the ileal mucosa and caecum compared with the ileal content of broiler chickens [30]. Although α-diversities of the JM and CC did not differ significantly, β-diversities showed significant separation between the JC, JM, and CC. This finding is in accordance with several others that compared microbial communities of the different GIT sections [31,32,33].

### 4.2. Composition of Gut Microbiota in the Different Sampling Places

As it has been described before, microbial communities were clearly separated in different sections of the GIT in broiler chickens [31,34,35,36]. In our study *Lactobacillus*, *Escherichia-Shigella*, *Streptococcus,* and *Corynebacterium* were the dominant genera in the JC. Most of the studies described how bacterial genera colonising the small intestine originate mainly from the phylum *Firmicutes* and include *Lactobacillus*, *Enterococcus*, *Streptococcus, Escherichia,* and *Clostridium* clusters [37,38,39,40]. However, the additional presence of *Bacteroides* and *Corynebacterium* was also reported [41,42,43].

In our case the major bacterial genera in the JM and CC were quite distinct from that of the JC. Anaerobes such as *Bacteroides*, *Ruminococcaceae UCG-014,* and *Faecalibacterium* and the aerotolerant anaerobic *Lactobacillus* showed dominancy in the JM and CC. In accordance with several other findings, the most abundant families within the caecum were *Clostridiaceae, Bacteroidaceae, Lactobacillaceae,* and butyrate producers such as *Lachnospiraceae and Ruminococcaceae* [17,32,42,44].

### 4.3. BW-Related Differences of the Gut Microbiota

In the JC, the genus *Enterococcus* (*Firmicutes*, *Bacilli* class) correlated positively with BW and showed significantly higher relative abundance in the HBW group. Members of the *Enterococcus* genus, especially *E. faecalis*, are generally considered to be harmful or even pathogenic. However, some species of *Enterococcus* are used as probiotics, improving gut health and having positive interactions with other members of the microbial community [44,45].

Based on correlation analysis, in the JM positive correlations were detected between body weight and genera *Alistipes*, *Bacteroides, Parabacteroides, Ruminococcaceae UCG-010,* and *Fournierella*. In the JM, only the genus *Coriobacteriaceae CHKCI002 (**Actinobacteria*) correlated with BW negatively. In the mucosa, *Parabacteroides*, *Alistipes*, and *Bacteroides* genera correlated with BW positively. The members of *Bacteroidetes* are adapted to gut mucosa using the mucin as substrate. The immune-modulatory effect, for example, of *B. fragilis,* is well known [46]. Their beneficial effects may explain their higher proportion in HBW chickens. Short chain fatty acids can modulate immune responses, and members of *Ruminococcaceae* are mostly butyrate producers [47]. From a human aspect, a high abundance of *Bacteroides* and *Ruminococcus* suggests a more healthy gut microbiota [48,49]. The *Ruminococcaceae* family is important for degrading pectin and cellulose in the colonic fermentation of dietary fibers [50]. Guo et al. found a similar diversity of bacterial communities in geese fed with different proportions of ryegrass. The ration of ryegrass affected the abundance of cellulose-degrading microbiota (*Ruminiclostridium* and *Ruminococcaceae UCG-010*) and enriched the lipid metabolic pathways [51]. *Bacteroides* maintain a complex and generally beneficial relationship with the host when retained in the gut [52]. *Parabacteroides* is a relatively new genus with distinctive features shared among other gut commensal bacteria [53]. Recently, studies on *P. distasonis* have displayed evidence that these bacteria are potentially beneficial [53]. *Alistipes* are commensal bacteria and their proportion increased with the age of broiler chickens [49]. There is contrasting evidence that *Alistipes* may have protective effects against some diseases, but other studies indicate *Alistipes* are pathogenic in humans [54]. Duggett et al. first isolated the genus *Coriobacteriaceae CHKCI002* from the chicken caeca and published its draft genome sequence [55]. To the best of our knowledge, there are no known reports of their occurrence in the ileum mucus of the chicken intestine. Their interaction with other microorganisms is unknown, but several species within the *Coriobacteriia* class have been implicated with human diseases [56,57].

In the CC, the relative abundance of *Alistipes* was significantly higher, whereas *Ruminococcaceae UCG-013* and *Negativibacillus* were lower in HBW chicken. In the CC, seven genera negatively correlated with BW, namely, *Defluviitaleaceae UCG-011*, *GCA-900066575*, *Butyricicoccus*, *Negativibacillus*, *Ruminiclostridium-9*, *Ruminococcaceae UCG-013,* and *Bilophila*. Potential performance-related phylotypes were assigned to some bacteria species such as *Lactobacillus salivarius, Lactobacillus aviarius, Lactobacillus crispatus, Clostridium lactatifermentans,* different members of the family Ruminococcaceae, *Bacteroides vulgatus*, *Akkermansia*, and *Faecalibacterium*, among others [2,3,58]. The relative abundances of four genera of *Ruminococcaceae* decreased in the HBW group. Ruminococci are generally beneficial key members of the gut ecosystem, having multiple interactions with the other members of gut microbiota [58].

Their decrease in the HBW group is probably due to the shift of the B/F ratio, which was relevant but not significant in the caecum. This is supported by the significant increase in proportion of *Alistipes,* a member of the phylum *Bacteroidetes* in the caecum.

In humans, Ley et al. and Turnbaugh et al. found lower relative abundance of *Bacteroidetes* and reduced biodiversity in the feces microbiota of obese individuals [59,60]. Other authors observed a higher relative abundance of *Bacteroidetes* in obese or overweight individuals compared with lean controls, and increased *Firmicutes* and *Actinobacteria* coupled with decreased *Proteobacteria* and *Fusobacteria* in obese individuals compared to normal-weight individuals [14,61,62].

The measured body weight-related differences in gut microbiota composition does not mean that bacteria are the determinant factor in the chicken’s growth potential. However, differences in some bacterial genera and species could generate complex humoral and immunological reactions. The exact mechanism of these bacteria–host interactions is not fully understood yet. On the other hand, the measured differences in the gut microbiota may also be the results of differences in the nutrient digestion of individual animals. Digestion affects not only the absorption rate of nutrients, but also the nutrients available for the microbes.

### 4.4. Analysis of Co-Occurrence Patterns

Co-occurrence interaction results in the JC revealed a correlation between the genera of *Actinobacteria* and *Firmicutes Bacilli* classes; however, their pattern was different in the LBW and HBW groups. More genera and interaction links were found in the LBW group. Similar to our results, Huang et al. have also found positive correlations between some bacteria belonging to the phylum *Actinobacteria* and to the class *Bacilli* in the foregut of chickens [61].

The genus *Corynebacterium* includes Gram-positive aerobic bacteria. They are widely distributed in nature in the gut microbiota of animals and humans and are mostly innocuous, most commonly existing in commensal relationships with their hosts [62,63]. Others can cause human disease, including most notably diphtheria by *C. diphtheriae* [63].

We found a single reference in the literature that described the interaction between *Corynebacterium* and the pathogenic species of *Staphylococcaceae* [64]. According to this research, the *Corynebacterium* species has a key role of as attenuator of *Staphylococcus aureus* virulence in the nose microbiota.

In the JM, correlation connections only in the LBW group were found. The network members included the genera of *Ruminococcaceae*, *Lachnospiraceae* families, *Christensenellaceae R-7 group*, and *Family XIII UCG-001* from the *Clostridia* class, and all had positive correlations. In our case in LBW chickens, *Lachnospiraceae* often showed positive correlations with *Ruminococcaceae*. Ruminococcus species are defined as strictly anaerobic, Gram-positive, non-motile cocci that do not produce endospores and require fermentable carbohydrates for growth [65]. According to Khoruts et al., *Ruminococcaceae* are one of the few types of bacteria involved in converting bile acids [66]. *Lachnospiraceae* belong to the core of gut microbiota, colonising the intestinal lumen from birth. Although members of Lachnospiraceae are among the main producers of short-chain fatty acids, different taxa of this family are also associated with different intra- and extraintestinal diseases in humans, such as ulcerative colitis, Crohn’s, and celiac disease [67]. Genomic analysis of *Lachnospiraceae* revealed a considerable capacity to utilise diet-derived polysaccharides, including starch, inulin, and arabinoxylan, with substantial variability among species and strains [67]. The seven-member node identified *Sellimonas* genus includes Gram-positive and anaerobic bacteria species previously considered as uncultivable. Although little is known about this *Lachnospiraceae* family member, its increased abundance has been reported in patients who recovered intestinal homeostasis after dysbiosis events [68].

As far as the JC-JM microbiota interaction is concerned, a significant number of correlations was found only in the LBW chickens. *Trichococcus* from the JC had positive correlations with seven genera from the JM. *Oligella* from the JC was also positively correlated with six genera from the JM. *Trichococcus* are lactic acid bacteria (LAB) that have the ability to utilise sugars, sugar alcohols, and polysaccharides [69]. It has already been identified from poultry feeds [70]. *Oligella* species are gram-negative organisms that cause infections primarily of the genitourinary tract. This pathogenic genus could be detected in JC samples only in LBW chickens. There is no information on these bacteria in poultry. *Oligella* species are described in humans with urological infections and diseases [71]. The mechanism of infection currently remains unclear. In the case of CC, no co-occurrence patterns have been found.

The explanation of these relationships is not easy. Our results highlight the need to investigate the composition and relationship of lumen and mucosa-linked microbiota in the small intestine. Whereas in the lumen the flux of nutrients is continuous, the consistent structure of mucosa provides a longer retention time and more networking chances for the bacterial community. The reason for the absence of bacterial co-occurrence in the caeca remains unclear.

In summary, beta diversity, according to canonical correspondence analysis, showed a significant separation of bacterial groups between the two BW categories. The BW-related changes in microbiota were mostly significant in the jejunal mucosa. The phylum *Bacteroidetes* was higher, whereas *Firmicutes* was lower in the JM of HBW chickens. At the genus level, *Enterococcus* in the JC and *Bacteriodes* and *Ruminococcaceae UCG-010* in the JM showed significantly higher abundance in the HBW animals. On the other hand, *Ruminococcaceae UCG-013* and *Negativibacillus* in the CC of LBW birds was higher. Similar co-occurrence patterns of LBW and HBW chickens were detected in the IC. In the JM, only LBW birds showed significant bacterial interaction between the genera of Firmicutes. Two genera, *Trichococcus* and *Oligella*, in the JC of LBW animals had co-occurrence connections with other genera in the JM. No such interactions in HBW chickens and in the CC of both BW groups have been found. According to our results, BW-dependent dominant and low abundant genera could modify the different metabolic pathways and, this way, the growth of the chicken.

## 5. Conclusions

This study confirmed that the composition of gut microbiota of chickens is influenced mostly by the sampling place. The digesta of the small intestine, the mucosa layer of the gut wall, and the caecal contents have microbiota with different diversities and compositions. The growth rate of birds depends on several nutritional and environmental factors and gut microbiota composition does not seem to be one of the strongest.

In general, caecal chymus and excreta are the most commonly studied sample types when examining the relationship between intestinal microbiota and the efficiency of poultry production. The novelty of this study is that we could prove that the increased *Bacteroidetes*/*Firmicutes* ratio in the jejunal mucosa could be a marker of the growth potential. Besides that, several mostly positive interactions have been detected between some genera in the JM and the growth rate of chickens. We could also detect two genera (*Trichococcus* and *Oligella)* in the IC of the LBW group that had an impact on the abundance of other bacterial genera in the mucosa. Further research is needed to obtain more of an understanding on how the mechanism works, how bacteria of the gut lumen can modulate the microbiota development of the mucosa, and how the microbes of mucosa can change the metabolism of the host by special metabolomic or gut-associated gene expression mechanisms.

## Figures and Tables

**Figure 1 animals-12-01296-f001:**
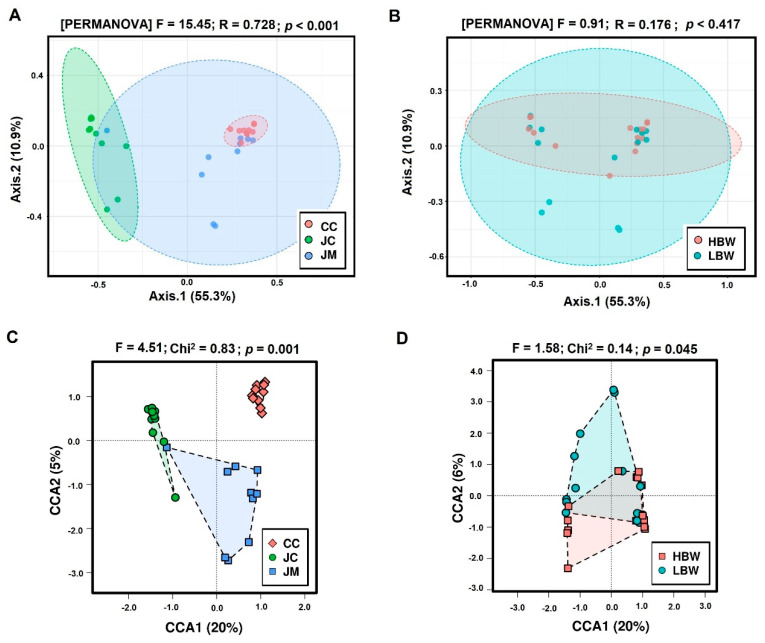
Principal coordinate analysis (PCoA) based on Bray–Curtis dissimilarity matrix on samples: (**A**) ordination of sampling places (JC, JM, and CC); (**B**) ordination of BW groups. Significance was examined with permutational multivariate analysis of variance (PERMANOVA) performed with 999 permutations. The differences were considered significant at a level of *p* < 0.05. (**C**) Canonical correspondence analysis (CCA) of gut microbiota clustering according to sample sites and (**D**) BW groups. Legend: JC—jejunum chymus in green; JM—jejunum mucosa in blue; CC—caecum chymus in red; LBW—low body weight; HBW—high body weight.

**Figure 2 animals-12-01296-f002:**
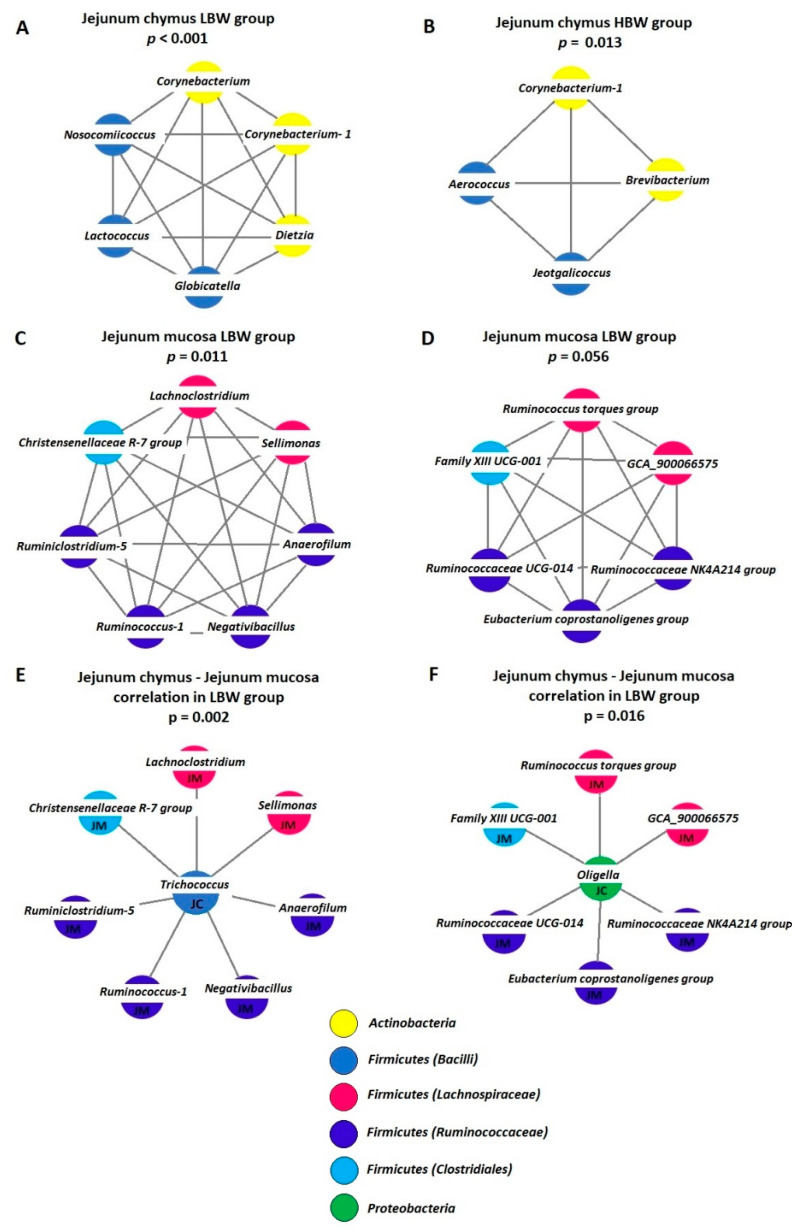
Co-occurrence patterns in the sampling places and changes in different body weights. (**A**) Genus correlations in jejunum chymus LBW; (**B**) jejunum chymus HBW; (**C**,**D**) jejunum mucosa LBW groups. (**E**,**F**) Jejunum chymus-jejunum mucosa connection in the LBW group.

**Table 1 animals-12-01296-t001:** Diversity indices of the intestinal microbiota of broiler chickens.

SP	BW	Diversity Indices (Mean)				
		OTU	Chao1	Shannon	Simpson	PD
JC	LBW	105	106	2.86	0.71	38.0
	HBW	122	122	3.17	0.72	41.8
JM	LBW	345	346	5.11	0.82	101.6
	HBW	360	361	6.45	0.97	109.2
CC	LBW	362	363	6.10	0.963	97.9
	HBW	350	351	5.98	0.955	97.5
JC		111 ^b^	111 ^b^	2.08 ^b^	0.71 ^b^	39.9 ^b^
JM		345 ^a^	346 ^a^	3.97 ^a^	0.89 ^a^	105.4 ^a^
CC		355 ^a^	357 ^a^	4.18 ^a^	0.96 ^a^	97.7 ^a^
LBW		269	270	3.24	0.83	79.2
HBW		272	272	3.59	0.88	82.9
Pooled SEM		23.67	23.74	0.31	0.03	6.25
*p*-Values	SP	<0.001	<0.001	<0.001	<0.001	<0.001
	BW	0.779	0.780	0.162	0.256	0.589
	SP × BW	0.860	0.863	0.247	0.311	0.891

^a,b^ Means in the same column with different letters differ significantly (*p* < 0.05). Legend: JC—jejunum chymus; JM—jejunum mucosa; CC—caecum chymus; OTU—operational taxonomic unit; PD—phylogenetic distance (whole tree).

**Table 2 animals-12-01296-t002:** Relative abundance of bacterial phyla in the different sampling places of broiler chickens as affected by BW (%).

	Body Weight	Sampling Place			Mean (BW)	FDR *p*-Values		
		Jejunal Chymus	Jejunal Mucosa	Caecum Chymus		SP	BW	SP × BW
*Actinobacteria*	LBW	2.54	1.55	0.38	**1.49**	0.188	0.898	0.681
	HBW	4.40	0.46	0.17	**1.67**			
	**Mean (SP)**	**3.47**	**1.00**	**0.28**				
*Bacteroidetes*	LBW	0.38	20.83 ^B^	45.25	**22.15 ^B^**	<0.001	0.029	0.109
	HBW	1.18	39.99 ^A^	51.61	**30.93 ^A^**			
	**Mean (SP)**	**0.78 ^c^**	**30.41 ^b^**	**48.43 ^a^**				
*Cyanobacteria*	LBW	4.82	0.47	0.05	**1.78**	0.163	0.990	0.878
	HBW	3.76	1.47	0.06	**1.76**			
	**Mean (SP)**	**4.29**	**0.97**	**0.06**				
*Deinococcus Thermus*	LBW	0.04	0.02	0.00	**0.02**	0.406	0.532	0.843
	HBW	0.02	0.00	0.00	**0.01**			
	**Mean (SP)**	**0.03**	**0.01**	**0.00**				
*Epsilonbacteraeota*	LBW	0.00	0.00	0.00	**0.00**	0.567	0.525	0.580
	HBW	0.01	0.11	0.00	**0.04**			
	**Mean (SP)**	**0.01**	**0.06**	**0.00**				
*Firmicutes*	LBW	88.16	74.76 ^A^	51.77	**71.57 ^A^**	<0.001	0.053 ^T^	0.532
	HBW	76.67	54.78 ^B^	46.92	**59.46 ^B^**			
	**Mean (SP)**	**82.42 ^a^**	**64.77 ^b^**	**49.35 ^c^**				
*Patescibacteria*	LBW	0.25	0.05	0.00	**0.1**	0.530	0.530	0.559
	HBW	4.03	0.12	0.00	**1.38**			
	**Mean (SP)**	**2.14**	**0.09**	**0.00**				
*Proteobacteria*	LBW	3.78	2.25	2.39	**2.81**	0.045	0.399	0.170
	HBW	9.88	2.72	0.98	**4.53**			
	**Mean (SP)**	**6.83 ^a^**	**2.48 ^b^**	**1.68 ^b^**				
*Tenericutes*	LBW	0.01	0.08 ^B^	0.15 ^B^	**0.08 ^B^**	0.056 ^T^	0.068 ^T^	0.406
	HBW	0.05	0.34 ^A^	0.25 ^A^	**0.21 ^A^**			
	**Mean (SP)**	**0.03 ^b^**	**0.21 ^a^**	**0.20 ^a^**				
B/F Ratio	LBW	0.00	0.32 ^B^	0.89	**0.40 ^B^**	<0.001	0.029	0.188
	HBW	0.02	0.76 ^A^	1.13	**0.64 ^A^**			
	**Mean (SP)**	**0.01 ^c^**	**0.54 ^b^**	**1.01 ^a^**				

Bacterial phylum differences between groups were assessed using a two-way ANOVA test, with Benjamini–Hochberg false discovery rate (FDR) correction. FDR-corrected *p*-values below 0.05 were considered as significant and results between 0.05 and 0.1 (0.05 < *p* < 0.10) were considered a trend (^T^). Body weight (BW) effects at each sampling place (SP) were also examined with a one-way ANOVA test and the significance of Tukey’s HSD post hoc test was indicated at *p* < 0.05. ^a,b,c^: values within the mean (SP) rows with different lowercase letters were significantly different (*p* < 0.05). ^A,B^: values within the mean columns with different capital letter superscripts were significantly different (*p* < 0.05).

**Table 3 animals-12-01296-t003:** Relative abundance of important bacterial genera in the different sampling places of broiler chickens as affected by BW (%).

Genus	Body Weight	Sampling Place			Mean (BW)	FDR *p*-Values		
		Jejunal Chymus	Jejunal Mucosa	Caecum Chymus		SP	BW	SP × BW
*Alistipes*	LBW	0.00	0.64	0.82 ^B^	**0.48 ^B^**			0.451
	HBW	0.03	1.71	2.48 ^A^	**1.41 ^A^**		0.081 ^T^	
	**Mean (SP)**	**0.01 ^b^**	**1.17 ^a^**	**1.65 ^a^**		0.017		
*Bacteroides*	LBW	0.11	19.06 ^B^	42.39	**20.52 ^B^**			0.280
	HBW	0.76	34.28 ^A^	46.27	**27.10 ^A^**		0.098 ^T^	
	**Mean (SP)**	**0.43 ^c^**	**26.67 ^b^**	**44.33 ^a^**		0.000		
*Enterococcus*	LBW	0.02 **^B^**	0.03	0.00	**0.02 ^B^**			0.001
	HBW	0.22 **^A^**	0.02	0.00	**0.08 ^A^**		0.012	
	**Mean (SP)**	**0.123 ^a^**	**0.025 ^b^**	**0.002 ^b^**		0.000		
*Ruminococcaceae* *UCG-010*	LBW	0.00	0.09 **^B^**	0.19	**0.09 ^B^**			0.472
	HBW	0.00	0.24 **^A^**	0.29	**0.18 ^A^**		0.083 ^T^	
	**Mean (SP)**	**0.00 ^b^**	**0.17 ^a^**	**0.24 ^a^**		0.000		
*Ruminococcaceae* *UCG-013*	LBW	0.01	0.09	0.39 **^A^**	**0.17 ^A^**			0.026
	HBW	0.01	0.06	0.18 **^B^**	**0.08 ^B^**		0.020	
	**Mean (SP)**	**0.010 ^b^**	**0.078 ^b^**	**0.285 ^a^**		0.000		

Bacterial genera differences between groups were assessed using a two-way ANOVA test, with Benjamini–Hochberg false discovery rate (FDR) correction. FDR-corrected *p*-values below 0.05 were considered as significant and results between 0.05 and 0.1 (0.05 < *p* < 0.10) were considered a trend (^T^). Body weight (BW) effects at each sampling place (SP) were also examined with a one-way ANOVA test and the significance of Tukey’s HSD post hoc test was indicated at *p* < 0.05. ^a,b,c^: values within the mean (SP) rows with different lowercase letters were significantly different (*p* < 0.05). ^A,B^: values within the mean columns with different capital letter superscripts were significantly different (*p* < 0.05).

## Data Availability

All data generated or analysed during this study are included in this published article (and its supplementary information files). Raw sequence data of the 16S rRNA metagenomics analysis are deposited in the National Center for Biotechnology Information (NCBI) Sequence Read Archive under the BioProject identifier PRJNA609272.

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
