# Peer review of "Microbiota Composition of Mucosa and Interactions between the Microbes of the Different Gut Segments Could Be a Factor to Modulate the Growth Rate of Broiler Chickens"

_animals, 2022, doi:10.3390/ani12101296_

Round 1

Reviewer 1 Report

Major revisions:

  1. In “Materials and Methods” part, lines155-162, considering both intestinal segments and body weight these two key factors, PERMANOVA analysis might be more suitable.
  2. The terrible quality of figures especially figure 3 influences the normal review.
  3. In “Materials and Methods” part “Bioinformatics and Statistical Analyses”, the description in quality control of origin data is missing.
  4. 4. Lines 164-169, the statistic analysis of microbial composition at different taxonomical levels and in different intestinal sampling places was performed using ANOVA test, it was of no sense, because microbial composition did not fit the normal distribution.

Minor revisions:

  1. Lines 28, 33, please use abbreviations correctly.
  2. In Figure 1,both P and R values are necessary.

3.In 3.2. Gut microbiota composition at phylum level, the statistical methods need to be re-checked.

  1. For example in table 2, figure 2, all taxonomic level names should be italicized, including on figures and in tables according to the guidelines of ASM.
  2. please improve the quality of figures (such as figure 2, figure 3).

Author Response

Reviewer 1:

Dear Reviewer!

Many thanks for reading through carefully the manuscript and giving suggestions for improving the quality of the paper. Please find the detailed answers below:

Major revisions:

Q1: In “Materials and Methods” part, lines155-162, considering both intestinal segments and body weight these two key factors, PERMANOVA analysis might be more suitable.

Thanks for the comments. The suggestion has been accepted and beta diversity evaluation was carried out with permutational multivariate analysis of variance. Figure 1 has been modified according to this.

Q2: The terrible quality of figures especially figure 3 influences the normal review.

We have taken Figure 2 out from the text and added to the supplementary materials file, because there was no possibility to enlarge this figure. All of the correlations of this figure can be found in the text. The quality and size of Figure 3 (now it is Figure 2) has been improved in the revised manuscript. We hope that you can accept it in this form.

Q3: In “Materials and Methods” part “Bioinformatics and Statistical Analyses”, the description in quality control of origin data is missing.

The missing information on the quality control of original data has been inserted in lines 147-149.

Q4: Lines 164-169, the statistic analysis of microbial composition at different taxonomical levels and in different intestinal sampling places was performed using ANOVA test, it was of no sense, because microbial composition did not fit the normal distribution.

The explanation why ANOVA was used for the comparison of the bacterial distribution has been described in the relevant Materials and Methods part in lines 175-179.

Minor revisions:

Q5: Lines 28, 33, please use abbreviations correctly.

Thanks for this remark. The full names of the abbreviations have been inserted.

Q6: In Figure 1,both P and R values are necessary.

Since the suggested PERMANOVA analysis was carried out, this figure has been changed and the new version contains the R, F and p values.

Q7: In 3.2. Gut microbiota composition at phylum level, the statistical methods need to be re-checked.

At Q4 above, we explain why ANOVA was used. Please accept this answer also at this point.

Q8: In table 2, figure 2, all taxonomic level names should be italicized, including on figures and in tables according to the guidelines of ASM.

It has been done in the mentioned table, figure and in the whole manuscript.

Q9: Please improve the quality of figures (such as figure 2, figure 3).

Sorry for the bad quality of these figures. They were in the pdf format still acceptable, but the quality in the word version was really bad. We have taken Figure 2 out from the text and added to the supplementary materials file, because there was no possibility to enlarge this figure. All of the correlations of this figure can be found in the text. The quality and size of Figure 3 (now it is Figure 2) has been improved in the revised manuscript. We hope that you can accept it in this form.

Reviewer 2 Report

The paper presented for review brings new elements to the current state of knowledge regarding the composition of the microflora of the mucous membrane and the interactions between microbes of different segments of the intestine may be a factor in modulating the growth rate of broiler chickens.

The purpose of the work is clearly stated. The conclusions of the conducted research are clear and result from the obtained research results.

The material used for the research is sufficient, the test methods have been selected accordingly.

The tables are clear. The differences between the different groups have been marked correctly. Some of the results are presented in the figures, but they are very small, especially figures No. 2 and 3. This makes it difficult to understand them. I suggest enlarging the figures.

Discussing the results against the background of other authors is very detailed.

The publications cited by the authors of the article are well selected. For the most part, the authors refer to the latest knowledge published in renowned scientific journals.

Author Response

Reviewer 2.

Dear Reviewer!

Many thanks for the positive evaluation of the paper.

Q1: Some of the results are presented in the figures, but they are very small, especially figures No. 2 and 3. This makes it difficult to understand them. I suggest enlarging the figures.

Regarding your comment on the quality of Figures 2 and 3, we have taken Figure 2 out from the text and added to the supplementary materials file, because there was no possibility to enlarge this figure. All of the correlations of this figure can be found in the text. The quality and size of Figure 3 (now it is Figure 2) has been improved in the revised manuscript. We hope that you can accept it in this form.

Reviewer 3 Report

Dear authors,

I congratulate you for your work. It is a very interesting work .

I have made some comments to further enhance your paper. Some adjustment need to be made.

The authors wrote about microbiota composition of jejunum mucosa and also jejunum and caecum chymo.

I fully agree with the authors on the choice of wanting to investigate any differences in the different tracts of the small intestine, as this is the most important portion from the point of view of nutrient absorption, interesting also the choice to investigate the possible difference in microbial population in the luminal content and not only in the mucosa.

The ileum is the longest segment, of the small intestine, harbors the majority of gut associated lymphoid tissue as the higher abundance of microbiota .It is well known that health and performance in a poultry are strongly influenced the modulation of immune response, in view of the above , why was not considered by the authors?

Regarding the main sections, kindly note as follows:

Introduction

Lane 61 the authors wrote “that several studies have attempted to identify how  the intestinal microbes associated with production traits”… add reference

Material and methods

2.2. Animal handling and sampling

The authors describe the total number of animals sampled, which was then randomly divided into the two groups with lower and higher weights, but do not describe how many members are in each individual group.

Lane 110 there is a typo, chickens is the interested word

Results

Nothing to note

Discussion

The study does’nt show substantial differences between luminal and mucosal content in the two groups LBW and HBW. what are your hypotheses?

Try to argue it better.

I hope that the suggestions can help you to improve your paper.

Best regards

Author Response

Reviewer 3.

Dear Reviewer!

Many thanks for reading through carefully the manuscript and make suggestions for improving the quality of the paper. Please find the detailed answers below:

Q1: It is well known that health and performance in a poultry are strongly influenced the modulation of immune response, in view of the above, why was not considered by the authors?

Since in this trial no immune stimulation has been done, no such feed additive was used, the immune response parameters of the chickens were not investigated. However, the reviewer is right, for the differences in the weight gain could be responsible at least partly the different immune status of the animals. Unfortunately, we cannot evaluate this factor at this stage of the research.

Q2: Lane 61 the authors wrote “that several studies have attempted to identify how the intestinal microbes associated with production traits”… add reference

Two such references have been added in line 61.

Q3: The authors describe the total number of animals sampled, which was then randomly divided into the two groups with lower and higher weights, but do not describe how many members are in each individual group.

Thanks for this remark. It was really missing this information from the Materials and methods part. Both the number of the two selected groups and the number of investigated animals has been added in lines 111-117.

Q4: Lane 110 there is a typo, chickens is the interested word

Thanks for the remark. It has been corrected.

Q5: The study doesn’t show substantial differences between luminal and mucosal content in the two groups LBW and HBW. What are your hypotheses? Try to argue it better.

We have inserted a few sentences to the end of the discussion of part 4.3. “BW Related Differences of the Gut Microbiota”. In this, we try to evaluate the responsibility of the measured bacteriota differences on the growth rate.

Round 2

Reviewer 1 Report

The manuscript can be accepted.